# TGT: Text-Grounded Trajectories for Locally Controlled Video Generation

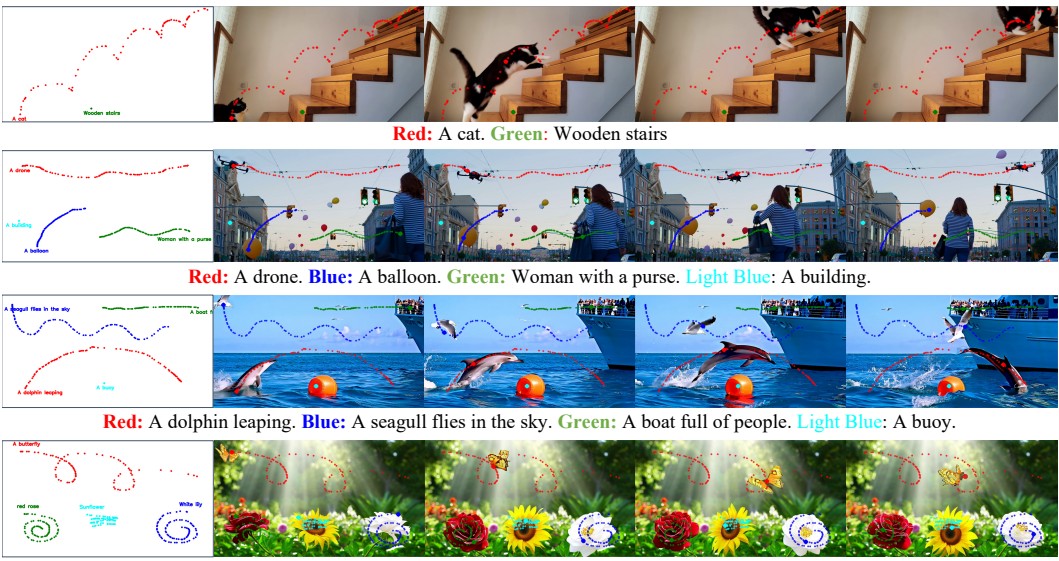

**Red:** A cat. **Green**: Wooden stairs

**Red:** A drone. **Blue:** A balloon. **Green:** Woman with a purse. **Light Blue:** A building.

**Red:** A dolphin leaping. **Blue:** A seagull flies in the sky. **Green:** A boat full of people. **Light Blue:** A buoy.

**Red:** A butterfly. **Blue:** White lily. **Green:** Red rose. **Light Blue:** Sunflower.

Figure 1: TGT generates videos following input trajectories with each trajectory associated with user specified local text prompt. Left: the user input trajectories or static points. Right: generated videos with trajectory visualizations, large dot indicate point location on the cut frame.

## Abstract

Text-to-video generation has advanced rapidly in visual fidelity, whereas standard methods still have limited ability to control the subject composition of generated scenes. Prior work shows that adding localized text control signals, such as bounding boxes or segmentation masks, can help. However, these methods struggle in complex scenarios and degrade in multi-object settings, offering limited precision and lacking a clear correspondence between individual trajectories and visual entities as the number of controllable objects increases. We introduce Text-Grounded Trajectories (TGT), a framework that conditions video generation on trajectories paired with localized text descriptions. We propose *Location-Aware Cross-Attention* (LACA) to integrate these signals and adopt a dual-CFG scheme to separately modulate local and global text guidance. In addition, we develop a data processing pipeline that produces trajectories with localized descriptions of tracked entities, and we annotate two million high quality video clips to train TGT. Together, these components enable TGT to use point trajectories as intuitive motion handles, pairing each trajectory with text to control both appearance and motion. Extensive experiments show that TGT achieves higher visual quality, more accurate text alignment, and improved motion controllability compared with prior approaches. Website: https://textgroundedtraj.github.io.

## 1 Introduction

Recent text-to-video models (Wang et al., 2025a; Brooks et al., 2024; Chen et al., 2023; Hong et al., 2023) achieve high visual fidelity and increasingly reliable prompt adherence. Despite substantial

progress driven by efforts to improve text responsiveness, prompts alone are a blunt instrument: they poorly specify spatial layout and motion (*e.g.*, where objects appear, their speed, and their trajectories). This limitation motivates a key question: how can we introduce explicit, fine-grained control into text-to-video generation while preserving realism and temporal coherence?

Recent works explore localized control in video generation along two main fronts. Structure-based controls enforce spatial layout using bounding boxes, edge maps, or segmentation masks (Ma et al., 2024b;a; Hu & Xu, 2023). While effective at preserving geometry, these signals are rigid and labor intensive. They require dense, frame-level supervision and are costly to author over long sequences, making them infeasible for direct user manipulation. Point-based trajectory controls, enabled by advances in point tracking (Doersch et al., 2023; 2024; Karaev et al., 2024b;a), offer a lighter alternative (Geng et al., 2025; Zhang et al., 2025; Namekata et al., 2025; Wang et al., 2025b). Here, users specify sparse 2D points that evolve over time. This paradigm achieves strong motion control in image-to-video (I2V), where the source image anchors identity and appearance. In text-to-video (T2V), however, the entity associated with each trajectory is not predetermined. The model must infer it from the caption, which often leads to ambiguous grounding, identity swaps, and off target motion when multiple entities are present. In short, current approaches either impose heavy supervision or leave the correspondence between entity and trajectory under specified in T2V setting.

Our method, Text-Grounded Trajectories (TGT), unifies the strengths of both lines of work. It uses sparse, point-based trajectories for flexible motion control while grounding a local text description to each trajectory to fix entity identity and appearance. Because no such dataset exists, we build a two-step data pipeline to create paired trajectory–text supervision. First, we finetune a vision language model to describe the entity at a given location $(x, y)$ in an image. We then segment entities in a specific frame of each video and sample a set of points per entity. Each point is then annotated with localized text using our distilled vision language model. Second, we propagate these points across frames with point tracking algorithms, producing full trajectories. Using this data pipeline, we create a large scale video dataset of trajectories paired with grounded text descriptions.

With generated data, we extend DiT-based T2V generation model with explicit grounding of motion and appearance. Concretely, we introduce Location-Aware Cross-Attention (LACA), a lightweight plug-in module added as an extra cross-attention branch in each DiT block. LACA aligns local text features with visual tokens near the trajectory, with gaussian weighting applied over space and time. Visual tokens not associated with any local text instead attend to the global video prompt. We finetune only the LACA modules, making them easy to integrate into pretrained backbones. At generation, we apply dual-CFG scheme with two separate classifier-free guidance scales: one for the global prompt and one for the grounded local text. This provides separate handles to preserve global semantics while enforcing localized control, thus enabling a flexible trade off between overall fidelity and motion precision. To our knowledge, we are the first to associate local text with point based trajectories for aligned video generation, introducing a new paradigm that enables more in depth control combining motion and entity. Quantitatively, TGT reduces trajectory error by nearly half compared to the strongest baseline, while maintaining the same level of video quality as the base model. In summary, our contributions can be summarized as follows:

- We propose TGT, a novel framework for controllable text-to-video generation. It introduces a lightweight plug-in module, Location-Aware Cross-Attention (LACA), together with a dual-CFG strategy. This design enables precise and disentangled control over both motion and appearance while remaining compatible with large scale pretrained DiT backbones.

- We design the first data collection pipeline to obtain paired trajectory–text supervision directly from raw videos, providing the missing supervision necessary for grounding local text in motion.

- Extensive experiments and human studies demonstrate that TGT achieves superior performance on visual quality, local grounding, and motion control compared to state-of-the-art baselines.

## 2 RELATED WORK

**Text-to-video generation.** Recent years have witnessed rapid progress in text-to-video (T2V) generation, driven by large-scale diffusion and transformer-based models (Wang et al., 2025a; Brooks et al., 2024; Chen et al., 2023; Hong et al., 2023; Kong et al., 2024; Ho et al., 2022a;b; Blattmann et al., 2023b;a; Xing et al., 2024; Chen et al., 2024; Yang et al., 2025; Lin et al., 2024). These meth-

ods excel in producing visually compelling clips that follow textual prompts, with improvements in temporal coherence and generalization to open-domain concepts. Despite this progress, most current models are limited prompt conditioning, which often fails to specify what should move, how motion should unfold, and where it should occur, whose motions are hard to be captured or correctly depicted by textual prompts. This motivates research into controllable video generation with explicit motion guidance that encloses mechanisms for spatial and temporal steering.

**Motion control in video generation.** Existing work seeks controllable video generation via motion guidance: structure-based methods impose spatial layouts with bounding boxes, segmentation masks, or edge maps, yielding precise alignment and sometimes constraining viewpoint or camera effects (Ma et al., 2024b;a; Hu & Xu, 2023), yet recent text-to-video models still lack strong control over how motion unfolds (Wang et al., 2025a; Brooks et al., 2024; Chen et al., 2023; Hong et al., 2023); moreover, these signals are rigid and labor-intensive, requiring dense frame-wise authoring over long sequences, while zero-shot approaches (Yu et al., 2024; 2023) reduce manual effort but often have weaker controllability, require extra motion planning (Su et al., 2023), or degrade motion. A complementary direction uses sparse point-based trajectories to indicate where and when motion should occur via 2D tracks, offering intuitive, fine-grained control (Doersch et al., 2023; 2024; Karaev et al., 2024b;a; Wang et al., 2025b; Wu et al., 2024; Geng et al., 2025; Zhang et al., 2025; Namekata et al., 2025); this excels in image-to-video, where the source image fixes identity and appearance, but in text-to-video the entity linked to a trajectory is not predetermined, so models must infer correspondence from the caption—causing ambiguous grounding, identity swaps, and off-target motion in multi-entity scenes. We address this gap by coupling trajectories with localized text to retain spatial alignment while improving motion accuracy across extended sequences.

# 3 METHOD

We build Text-Grounded Trajectories (TGT) on top of pretrained DiT-based video generation backbones, in particular Wan2.1 (Wang et al., 2025a). To enable text-grounded trajectory control while preserving the scalability of the backbone, we introduce Location-Aware Cross-Attention (LACA) (Section 3.2). We further develop a local text aware training and inference pipeline (Section 3.3) together with a data collection pipeline (Section 3.4). An overview of TGT is shown in Figure 2.

## 3.1 PRELIMINARIES

**Diffusion Transformer (DiT).** Diffusion models learn a generative process by progressively denoising a sample through a sequence of timesteps. Given a clean video $X_0$, the process follows:

$$q(X_t \mid X_{t-1}) = \mathcal{N}\big(\sqrt{1 - \beta_t}\, X_{t-1},\, \beta_t \mathbf{I}\big), \tag{1}$$

where $\{\beta_t\}_{t=1}^{T}$ is the variance schedule and $\{X_t\}_{t=1}^{T}$ is the noisy sequence. The denoiser $\epsilon_\theta$ is trained to predict the injected noise $\epsilon$ under condition $\mathcal{C}$, where $\mathcal{C}$ is a set of conditioning signals that may include text prompts, or other modalities, etc. The training objective is then defined as

$$\mathcal{L} = \mathbb{E}_{X_0, t, \epsilon}\left[\big\|\epsilon - \epsilon_\theta(X_t, t, \mathcal{C})\big\|_2^2\right]. \tag{2}$$

Unlike U-Net based backbones, DiTs (Peebles & Xie, 2023) replace convolutional blocks with transformer layers (Vaswani et al., 2017) that operate on spatio-temporal tokens. This design improves scalability for high dimensional video and enables seamless integration with long language features.

**Text-conditioned video generation.** In DiT–based text-to-video models, a text prompt $c$ that describes the video is embedded by a language encoder $\Phi(\cdot)$ into global text features $F_{\text{glob}} = \Phi(c) \in \mathbb{R}^{L \times D}$. The video is represented in a latent space and patchified into a token sequence $Z \in \mathbb{R}^{N \times D}$ with positional encodings, and diffusion timestep embeddings are incorporated into hidden states. Each DiT block is composed of self-attention over $Z$ to capture long-range space–time structure and text–video cross-attention, which injects semantic guidance, $F_{\text{glob}}$, into the visual tokens. We use standard multi-head cross-attention with $M$ heads and per-head width $D_h = D/M$:

$$H^{(m)} = \text{softmax}\left(\frac{Q^{(m)} K^{(m)\top}}{\sqrt{D_h}}\right) V^{(m)} \in \mathbb{R}^{N \times D_h}, \tag{3}$$

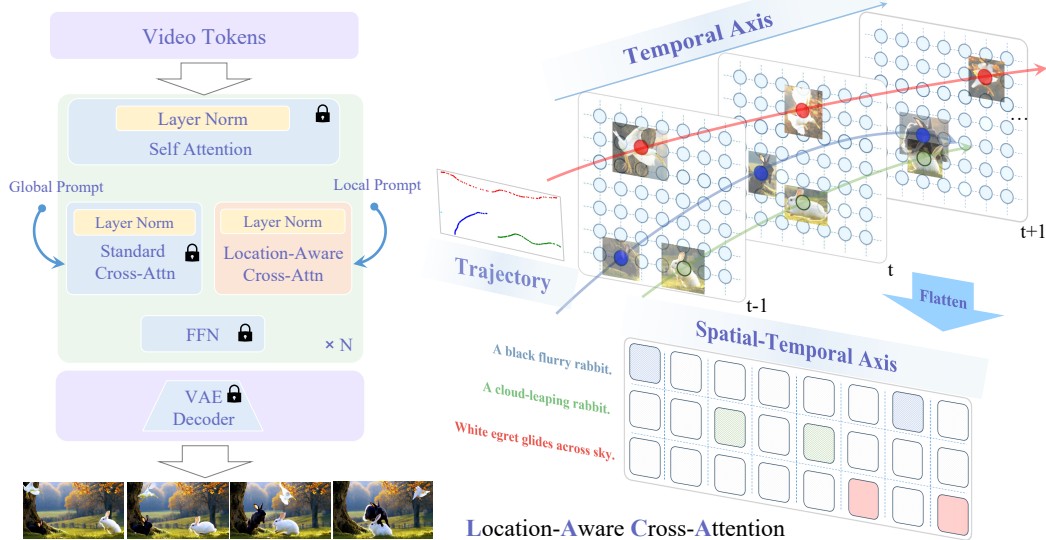

Figure 2: Pipeline of TGT. We propose Location-Aware Cross-Attention (LACA), a cross-attention that injects entity and location information into every DiT block's forward pass. LACA performs masked attention, ensuring each visual token attends only to its corresponding local text token.

Where $H^{(m)}$ denotes the output of the $m$-th attention head, representing a text-conditioned refinement of the input video tokens. By concatenating the outputs from all $M$ heads, we obtain

$$\text{CrossAttn}(Z, F_{\text{glob}}) = \left[ H^{(1)} \parallel \cdots \parallel H^{(M)} \right] W_O \in \mathbb{R}^{N \times D}, \tag{4}$$

where $W_O \in \mathbb{R}^{(MD_h) \times D}$ is a learnable MLP projection. This cross-attention enforces global semantic alignment to the text condition $c$ across video tokens. While effective for providing global semantic control, purely global cross-attention often overlooks fine-grained spatial details, motivating the localized conditioning mechanisms introduced in the following section.

**Classifier-Free Guidance (CFG).** To modulate the strength of the text condition during video generation, we apply the CFG (Ho & Salimans, 2022). Given a condition $c$, conditional prediction $\epsilon_\theta^{\text{cond}} = \epsilon_\theta(x_t, t, c)$ and unconditional prediction $\epsilon_\theta^{\text{uncond}} = \epsilon_\theta(x_t, t, \varnothing)$, the guided prediction is

$$\hat{\epsilon} = \epsilon_\theta^{\text{uncond}} + s \cdot \left( \epsilon_\theta^{\text{cond}} - \epsilon_\theta^{\text{uncond}} \right), \tag{5}$$

where $s$ is the guidance scale. A larger value of $s$ enforces stronger adherence to the condition $c$.

### 3.2 LOCATION-AWARE CROSS-ATTENTION

**Tokenization and backbone.** Let the denoising latent volume be $X \in \mathbb{R}^{T \times H \times W \times C}$. After patchification, we obtain a token sequence $Z \in \mathbb{R}^{L \times D}$, where $L = \frac{THW}{s}$ for a patch factor $s$, and $D$ is the embedding dimension. In parallel, global and local texts are encoded with the standard text tokenizer used for T2V. Each DiT block then applies (i) self-attention over $Z$ to capture spatio-temporal dependencies, (ii) cross-attention to inject the text features, and (iii) timestep based modulation.

**Text-grounded trajectories.** To inject text-grounded trajectory control into the base model, we introduce an additional location-aware cross-attention (LACA) branch. We first align each trajectory to the latent space. A trajectory is then defined as $\mathcal{T} = \{(p_t, m)\}_{t=1}^T$, where $p_t = (x_t, y_t, v_t)$ contains the 2D coordinate $(x_t, y_t)$ and a visibility flag $v_t \in \{0, 1\}$. $m$ is the local textual describing the object moving along this trajectory $\mathcal{T}$. To inject localized semantics, we map the text $m$ to a neighborhood $\mathcal{B}_r(x_t, y_t)$ of radius $r$ centered at $(x_t, y_t)$ at frame $t$, weighted by a Gaussian kernel

$$G_t(i, j) = \exp\left( -\frac{(i - x_t)^2 + (j - y_t)^2}{2\sigma^2} \right), \quad (i, j) \in \mathcal{B}_r(x_t, y_t). \tag{6}$$

The localized text feature, broadcast over spatial tokens in neighborhood, is then defined as:

$$F_t(i, j) = G_t(i, j) F_m, \qquad F_m = \Phi(m) \in \mathbb{R}^{L \times D}. \tag{7}$$

Figure 3: TGT data collection pipeline. We first apply Grounded SAM to segment entities and select representative points on every entity. These points are then passed to a point tracker and a distilled VLM to extract trajectories and localized captions, yielding paired trajectory–text supervision.

**Location-Aware Cross-Attention (LACA).** We define the cross-attention source $h_{t,ij}$ for each video token $z_{t,ij}$ as follows: if a trajectory point at time $t$ is visible ($v_t = 1$) and the token lies within the local neighborhood $\mathcal{B}_r(x_t, y_t)$ around the trajectory location, then $h_{t,ij}$ is set to the localized text feature $F_t(i, j)$. Otherwise, the token attends to the global caption feature $F_{\text{glob}}$:

$$h_{t,ij} = \begin{cases} F_t(i, j), & \text{if } v_t = 1 \text{ and } (i, j) \in \mathcal{B}_r(x_t, y_t), \\ F_{\text{glob}}, & \text{otherwise.} \end{cases} \tag{8}$$

Let $Q'$, $K'$, $V'$ be learnable projections (applied with multi-head in practice). The LACA update is

$$H(z_{t,ij}) = \text{softmax}\left( \frac{Q'(z_{t,ij})K'(h_{t,ij})^\top}{\sqrt{D}} \right) V'(h_{t,ij}). \tag{9}$$

Thus, when the point is visible, the tokens within $\mathcal{B}_r(x_t, y_t)$ attend to the local description of the trajectory, and all other tokens attend to the global prompt. In this way, LACA injects spatially targeted entity and motion while maintaining adherence to global prompt. Noticeably, LACA is a lightweight plugin branch, enabling controllable video generation without modifying the DiT backbone.

## 3.3 TRAINING AND INFERENCE

**Training objective.** During training, we only optimize the LACA module while keeping other parameters of the pretrained model fixed. We follow a flow-matching objective with velocity prediction for our loss. Let $X_1$ denote ground-truth video latent and $X_0 \sim \mathcal{N}(0, 1)$ denote Gaussian noise. Then for a timestep $t$, we have $X_t = tX_1 + (1 - t)X_0$, and the model $v_\theta$ predicts velocity $V_t = \frac{\partial}{\partial t} X_t = X_1 - X_0$. Then our training objective is formulated as:

$$\mathcal{L}(\theta) = \mathbb{E}_{t, X_0, X_1} \left[ \left\| V_t - v_\theta(X_t, t \mid \mathcal{C}) \right\|_2^2 \right], \tag{10}$$

where $\mathcal{C}$ is the union of all conditions, including video prompts and text-grounded trajectories.

**Conditioned video generation.** To control text condition guidance during generation, we adopt a dual-CFG strategy. We apply independent dropout on global and local text to adopt separate guidance scales during generation. This enables flexible control between overall prompt adherence and text-grounded trajectory control. Given conditional and unconditional predictions:

$$\epsilon^{\text{none}} = \epsilon_\theta(x_t, t, \varnothing), \ \epsilon^{\text{glob}} = \epsilon_\theta(x_t, t, c_{\text{glob}}), \ \epsilon^{\text{loc}} = \epsilon_\theta(x_t, t, c_{\text{loc}}), \ \epsilon^{\text{both}} = \epsilon_\theta(x_t, t, c_{\text{glob}}, c_{\text{loc}}), \tag{11}$$

and the global and local guidance scale $s_{\text{glob}}$ and $s_{\text{loc}}$, the guided prediction is then computed as:

$$\hat{\epsilon} = \epsilon^{\text{none}} + s_{\text{glob}}\left(\epsilon^{\text{both}} - \epsilon^{\text{glob}}\right) + s_{\text{loc}}\left(\epsilon^{\text{both}} - \epsilon^{\text{loc}}\right). \tag{12}$$

Moreover, since global and local text cross-attention are implemented as separate branch, we can explicitly balance their relative influence during inference. Formally, the hidden update is

$$Z^{\text{next}} = (1 - \lambda) \cdot \text{CrossAttn}(Q, K, V) + \lambda \cdot \text{LACA}(Q', K', V'), \qquad (13)$$

where $\lambda$ is a hyperparameter. The model degenerates to a standard text-to-video generator when $\lambda = 0$, while a nonzero $\lambda$ allows explicit balancing between global and local guidance.

### 3.4 LABEL LOCAL TEXT AND MOTION

Due to the lack of trajectories associated with text, we propose a data pipeline that constructs text-grounded trajectories from raw videos. The overall process is illustrated in Figure 3.

**Video captioning.** For each raw video, we employ Qwen2.5-VL (Bai et al., 2025) to generate a global textual description that captures the overall scene and context.

**Local text labeling.** To obtain localized semantic labels, we adopt a teacher–student strategy with vision language models (VLMs). Specifically, given an image from the COCO dataset (Lin et al., 2014) and a spatial coordinate $(x, y)$, we draw a small circle at the designated location and prompt GPT-4o (Achiam et al., 2023) to describe the entity at that point (*e.g.*, "a man riding a bicycle," "a yellow traffic light"). This process yields a triplet of image, point, and text. We then use these annotations to finetune Qwen2.5VL-3B (Bai et al., 2025), training it to accept an image together with a coordinate-based query such as "What is the item at location (x,y)?" and return the corresponding localized description. After finetuning, Qwen2.5VL-3B can generate accurate entity-level annotations conditioned on spatial coordinates, without requiring visual markings on the image. This distillation process transfers the descriptive ability of the teacher model to a lighter model suited for large-scale data construction. More details about the model are in Appendix A.1.

Then we apply this finetuned Qwen2.5VL-3B to generate localized captions on a raw video frame. Specifically, we first run Grounded SAM (Kirillov et al., 2023; Ravi et al., 2024) on a specific frame to obtain entity masks. We then sample representative points on each entity based on the size of each entity mask. Each sampled representative point paired with a coordinate based query is fed into our finetuned Qwen2.5VL-3B, which returns a localized description of the corresponding entity. In this way, every entity in the scene is linked to one or more annotated anchor points, establishing the semantic grounding needed for the subsequent trajectory construction stage (details Appendix A.2).

**Object tracking.** In the final stage, we convert static point–text annotations into trajectories. Using Tracking-Any-Point (TAP) (Doersch et al., 2023), we propagate the sampled points across the subsequent frames of the video. Each trajectory maintains its association with the original localized text, providing both motion information and semantic grounding over time. Visibility flags are also recorded, allowing the trajectory to encode cases of occlusion or when an entity moves out of frame. The result is a set of temporally consistent trajectories, each paired with descriptive text, that captures both where an entity is located and what it is throughout the video (details Appendix A.2).

## 4 EXPERIMENTS

We comprehensively evaluate TGT across standard benchmarks and real-world scenarios. Section 4.1 details datasets, training protocols, evaluation metrices, and baselines. Section 4.2 reports quantitative metrics, user study results, and qualitative comparisons versus baselines. Section 4.3 demonstrates several applications, such as video-to-video generation, where dense motion trajectories and text are extracted from a source video to guide the synthesis of new or edited content. Finally, Section 4.4 presents ablation studies that isolate the contribution of each component.

### 4.1 EXPERIMENTAL SETUP

**Dataset.** We constructed our training set on a large scale internal dataset by screening five million high quality video clips. After removing scene cuts and enforcing strict aesthetic and motion standards, we curate a final subset of 2.4 million clips that contain strong, sustained object motion. For quantitaive evaluations, we adopt the publicly available DAVIS (Pont-Tuset et al., 2017) dataset. Following prior practice, we extract the first frame of each video and utilize the provided ground-truth segmentation mask as the standard input representation. This mask allows us to derive center

Table 1: Comparison of different methods on global/local CLIP-T and EPE. Best scores are in **bold**, second best are underlined. Wan denotes the Wan2.2 14B T2V model (Wang et al., 2025a).

| Method | CLIP-T ↑ (Global) | CLIP-T ↑ (Local) | EPE ↓ |
|---|---|---|---|
| Wan (Global) | **0.3408** | 0.2308 | 265.03 |
| Wan (Global + Local) | 0.3309 | 0.2394 | 180.36 |
| MotionCtrl | 0.3186 | 0.2291 | 74.33 |
| TrailBlazier | 0.3145 | 0.2408 | 65.15 |
| Tora | 0.3288 | 0.2423 | 47.41 |
| **Ours** | 0.3314 | **0.2531** | **25.11** |

Table 2: User study results on GSB preference (ours vs. baseline). Positive values indicate preference for ours, negative values indicate preference for baselines. Larger magnitudes reflect stronger preference.

| Method | Visual Quality | Motion Control | Prompt Control |
|---|---|---|---|
| Wan (Global + Local) | -35.0 | 65.0 | 51.7 |
| MotionCtrl | 96.7 | 61.7 | 68.3 |
| TrailBlazier | 98.3 | 78.3 | 81.7 |
| Tora | 73.3 | 38.3 | 38.3 |

point tracks and to compute bounding boxes required by certain baselines. On average, we have approximately $2 \sim 3$ points tracks or bounding boxes per video in our test set.

**Evaluation metrics.** We report metrics that evaluate both quality and controllability of generated videos. For semantic alignment, we adopt CLIP-T scores at global and local levels. Global CLIP-T measures overall consistency between generated video and video prompt, while local CLIP-T evaluates alignment between local text prompts and the corresponding regions. We crop a window with radius $R = \tau \cdot \min(H, W)$, where $\tau \in \{0.05, 0.10, 0.15, 0.20\}$, and compute CLIP-T between local text and the video in that window. The final local CLIP-T score is averaged across windows of different sizes. Motion controllability is measured by End-Point Error (EPE), the L2 distance between the condition tracks and the trajectories estimated from generated videos.

**Implementation details.** We implement TGT based on the Wan2.1 T2V 14B model (Wang et al., 2025a). We fine-tune only the LACA branch on 48 H100 GPUs in two stages: dense trajectories ($\sim 40$ tracks per video) without applying Gaussian weighting or neighborhood constraints, followed by sparse trajectories ($\leq 5$ tracks) with gaussian kernels of $\sigma = 1$ and a neighborhood range of $r = 2$ for $200K$ steps. We use the AdamW optimizer ($\beta_1 = 0.9$, $\beta_2 = 0.999$) with a weight decay of $0.01$. The learning rate is set to $1 \times 10^{-5}$, and we clip the gradients at $10.0$. We set dropout rates to be $0.8$ for global prompts and $0.1$ for local texts during training. Both training and generation are conducted on videos of $832 \times 480$ resolution, 81 frames, and 16 fps. At generation, we apply our dual-CFG strategy with global and local guidance scales of 5 and 4, balanced by $\lambda = 0.5$.

**Baselines.** We compare our method with 5 strong baselines, including a large scale T2V generation model, WanT2V 2.2 14B (Wang et al., 2025a), evaluated both with and without extended video prompts that describe motion and location (details in Appendix B.1), a bounding box based controllable video generation approach, TrailBlazer (Ma et al., 2024b) (details in Appendix B.3), and two trajectory based methods, MotionCtrl (Wang et al., 2024) and Tora (Zhang et al., 2025) (details in Appendix B.2). All baselines take the **same** motion and text from inputs, except WanT2V 2.2 (global + local) takes some additional global prompt describing motion and location.

## 4.2 EXPERIMENTAL RESULTS

**Quantitative comparison.** Table 1 shows that TGT achieves strong controllability while preserving the high quality generation capacity of large scale T2V models. Although WanT2V with extended prompts can incorporate high level motion and location descriptions, such prompts remain insufficient for fine grained control, as reflected in its large EPE error. TrailBlazer, which generates localized representations and later fuses them into a complete video, suffers from degraded video quality. Our method demonstrates clear superiority over trajectory-based methods as well. TGT nearly halves the end point error relative to the strongest baseline (Tora) while also achieving the highest local CLIP-T score. Overall, TGT produces the most controllable videos among all baselines, while maintaining a comparable level of visual quality compared to the base T2V model.

**User study**. Table 2 quantifies human preference for comparison of generated video from ours versus baselines. Each comparison is labeled as one of three outcomes: $G$ (ours preferred), $S$ (no clear preference / indistinguishable), or $B$ (baseline preferred). Let $G$, $S$, and $B$ denote the respective

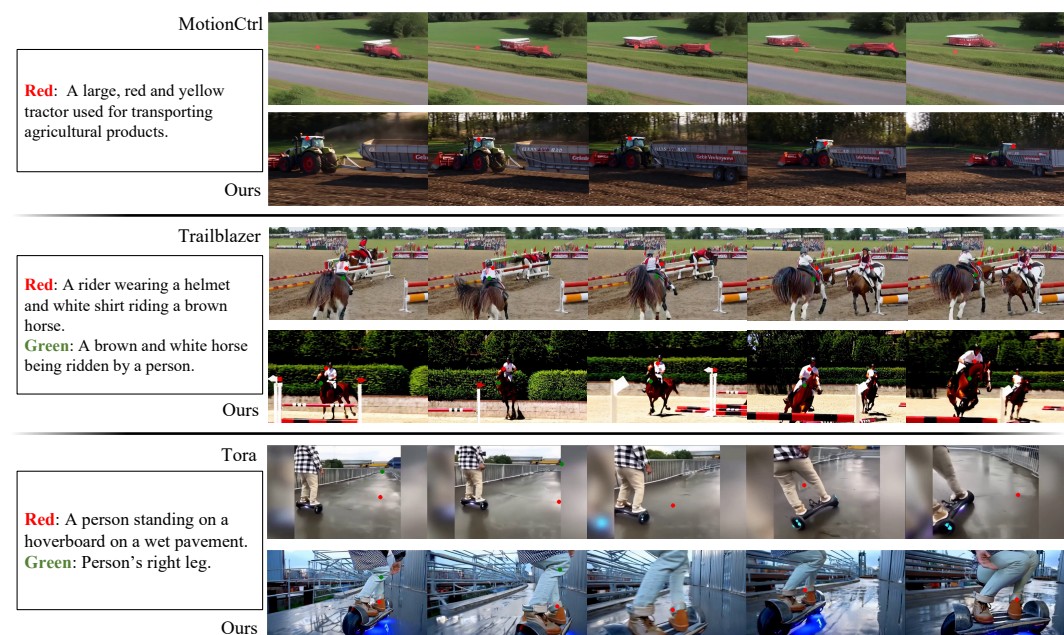

Figure 4: Qualitative comparison between TGT and baseline methods. Input point locations are labeled in color on each video frame. Local text prompts are shown on the left. Our method generates more aligned and realistic entity that follows the local text and achieves greater precision in motion.

counts over all comparisons. We define the GSB percentage as $100 \times \frac{G-B}{G+S+B} \in [-100, 100]$. A positive value indicates that users, on balance, prefer our method over the baseline while a negative value indicates the opposite. 0 corresponds to parity once "no preference" votes are taken into account. The magnitude reflects the strength of the preference where larger absolute values occur when one method receives substantially more wins than the other. Noticeably, a higher proportion of $S$ votes reduces the magnitude by increasing the normalization term.

**Qualitative comparison.** We present qualitative results on DAVIS, comparing our method with baselines across multiple examples (Fig. 4). TrailBlazer attains reasonable grounding, but its generate-and-fuse strategy introduces temporal artifacts and object discontinuities. MotionCtrl and Tora improve stability yet still exhibit motion drift and weak alignment between local prompts and target regions in T2V. In contrast, our method produces semantically faithful videos where objects follow trajectories with strong temporal coherence. For example, the tractor and rider motion appears sharper and smoother, while the hoverboard case shows precise spatial grounding.

**Qualitative results from user input.** Figure 1 and Appendix C show TGT delivering strong control in complex scenarios, yielding prompt-faithful, aesthetically pleasing videos with rare distortion.

### 4.3 PRACTICAL APPLICATIONS

Beyond our main experiments, TGT naturally enables two interesting practical applications. While not our primary objective, these examples highlight the potential of TGT to provide precise control for faithful video-to-video reconstruction as well as flexible, targeted, text-guided edits.

**Video-to-video mirroring.** Given an input clip, we extract dense trajectories and localized text descriptions, then generate a mirrored video with TGT. Figure 5a shows that the generated sequence faithfully preserves the subject's smooth hand motion and the camera shifts, bringing the windows into view behind him, while also maintaining correct grounding of surrounding items.

**Text-driven local editing.** By making small edits to the global and localized text (*e.g.*, replacing "man/person" with "werewolf"), TGT produces a modified video that keeps the original motion and layout while changing identity and appearance. In Figure 5b, our method maintains the man's movement and also controls scene attributes such as the *sofa*, *plant*, *calculator*, *table*, and *T-shirt* via local prompts, even when these details are not specified in the global video prompt.

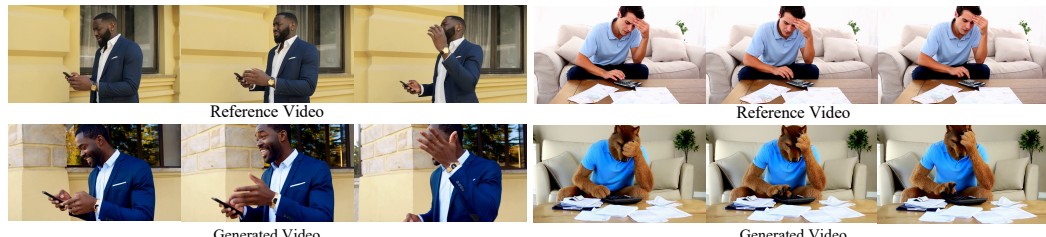

(a) Video-to-video mirroring          (b) Text-driven local editing.

Figure 5: Practical applications of TGT. We extract dense motion trajectories and local captions from a reference video and use them as conditions to generate a new video. The left example shows video-to-video mirroring, while the right demonstrates text-driven local editing.

Table 3: Ablation on components of LACA.

| Method | CLIP-T ↑ (Global) | CLIP-T ↑ (Local) | EPE ↓ |
|---|---|---|---|
| Dense track | 0.3307 | 0.2394 | 58.01 |
| + Sparse track | 0.3312 | 0.2447 | 45.28 |
| + Gaussian masking | **0.3314** | **0.2527** | **25.11** |

Table 4: Ablation on different CFG settings.

| Method | CLIP-T ↑ (Global) | CLIP-T ↑ (Local) | EPE ↓ |
|---|---|---|---|
| Only global | 0.3297 | 0.2480 | 91.38 |
| Only local | 0.3117 | 0.2493 | 43.29 |
| Global w/ local | 0.3307 | 0.2491 | 53.30 |
| Dual-CFG | **0.3314** | **0.2527** | **25.11** |

## 4.4 ABLATION STUDY

**Effectiveness of LACA with Gaussian Setup.** We assess LACA under the gaussian setup via an incremental ablation: start with dense tracks, add sparse-track tuning, then introduce gaussian modeling, isolating each component's effect. As shown in Tab. 3, moving from dense only to sparse track tuning yields small but consistent gains in global and local CLIP-T, while cutting EPE by roughly 25 percent. Adding gaussian masking delivers the largest jump where we have best performance across all four metrics, and EPE drops to about half of the dense only baseline. Overall, these results demonstrate that LACA with gaussian modeling enhances quality of generated video, increases alignment between text and video, and substantially improves trajectory accuracy.

**Dual-CFG Strategy.** We further ablate our conditional guidance strategy by comparing four different configurations of classifier-free guidance (CFG): only on global prompts, only on local prompts, treating global and local as combined conditions, formulated as $\hat{\epsilon} = \epsilon^{\text{none}} + s \cdot \left( \epsilon^{\text{both}} - \epsilon^{\text{none}} \right)$, and our proposed dual-CFG scheme with separate scales. From Tab. 4, only global guidance favors video quality but suffers in local alignment and has the largest EPE. Only local improves EPE substantially and slightly helps local alignment, but it degrades global quality significantly. Combing global and local lands in between. In contrast, our dual-CFG scheme achieves the best global and local CLIP-T simultaneously, and attains the lowest EPE by a wide margin. Decoupling the guidance strengths therefore provides a better balance between appearance alignment and trajectory stability.

## 5 CONCLUSION

We propose Text-Grounded Trajectories (TGT), a framework that conditions point trajectories paired with localized text on T2V generation. We design Location-Aware Cross-Attention (LACA) to integrate text-grounded trajectories into standard T2V model, and adopt dual-CFG scheme in generation. To support TGT, we design a scalable data pipeline and curate a large corpus of two million high-quality clips for training. Together, these components enable fine-grained, text-grounded control of appearance and motion in complex, multi-object scenarios while preserving visual fidelity and temporal coherence. Extensive experiments confirm that TGT delivers substantial improvements over concurrent approaches while maintaining state-of-the-art visual quality. Human studies further validate its effectiveness in real-world settings. Beyond quantitative gains, we highlight two practical applications, including faithful video-to-video reconstruction and targeted local editing, that showcase the versatility of our framework. Looking forward, we plan to extend TGT to broader settings such as more complex video-to-video editing or local-text-controlled image-to-video generation, aiming to push the boundary of semantically aligned, controllable video generation.

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

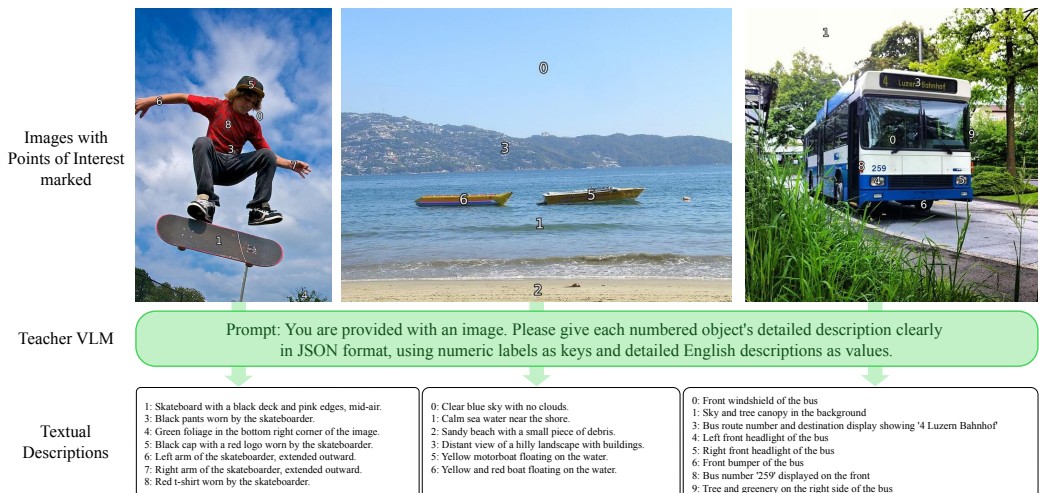

Figure 6: Data construction for distilling the local point caption VLM from an standard large VLM model. As the image shown, we prompt the teach model to generate labels via superimpose numbers onto images. Thereby the teacher model generate textual descriptions based on the numbered images.

# A    LABEL LOCAL TEXT AND MOTION

## A.1    POINT DESCRIPTION VLM

We adopt a teacher-student strategy to train a lightweight vision-language model capable of generating textual descriptions for arbitrary points within an image.

**Training set construction.** Following a procedure similar to (Yang et al., 2023), we build point-conditioned captioning supervision on the train 2017 split of COCO (Lin et al., 2014) dataset by first extracting candidate points of interest (POIs) from each image; the supervision signal is provided as localized captions paired with specific image coordinates. Concretely, we run Ultralytics SAM2.1-large (Ravi et al., 2024) to obtain instance masks and compute the geometric centroid of every mask in image coordinates. To avoid redundant, tightly clustered centroids, we perform non-maximum suppression (NMS) in point space by treating each centroid as a disk of fixed radius $r = 32$ pixels and randomly retaining a single representative point when overlaps occur. The resulting set of de-duplicated centroids constitutes the POIs for that image.

For annotation, we assign each POI a unique integer identifier and render the identifiers onto the image for reference (see Figure 6). The annotated image is then provided to a teacher VLM (GPT-4o in our implementation), which is prompted to produce a concise localized description for every identifier, yielding one text string per point. Each supervision unit is recorded as an $\langle \text{image}, (x, y), \text{text} \rangle$ triplet, where $(x, y)$ denotes the POI's absolute pixel coordinates with the image's native resolution. Collecting such triplets over the full split produces a large corpus of image–point–text pairs that we subsequently use to train the student model to generate local text given an image and a coordinate query.

**Training details.** We train a Qwen2.5-VL-3B model as the student model to map an input image and a pixel-coordinate query to a localized caption using the image-point-text triplets described above (see Figure 7). The training prompt is a single-line instruction: "`<image>Generate a detailed caption for the object at (x, y)`". No visual markers are rendered on the image. We freeze the vision encoder and update the multimodal projection module and the language model decoder. Optimization uses AdamW (Loshchilov & Hutter, 2019) with learning rate $2\times10^{-6}$, cosine decay, warmup ratio $0.03$, weight decay $0$, gradient clipping at $1.0$, and `BF16` precision. We train for 5 epochs with per-device batch size $= 4$ and gradient accumulation $= 4$ on 16 NVIDIA A100 GPUs; the global batch size per update is $4 \times 4 \times 16 = 256$.

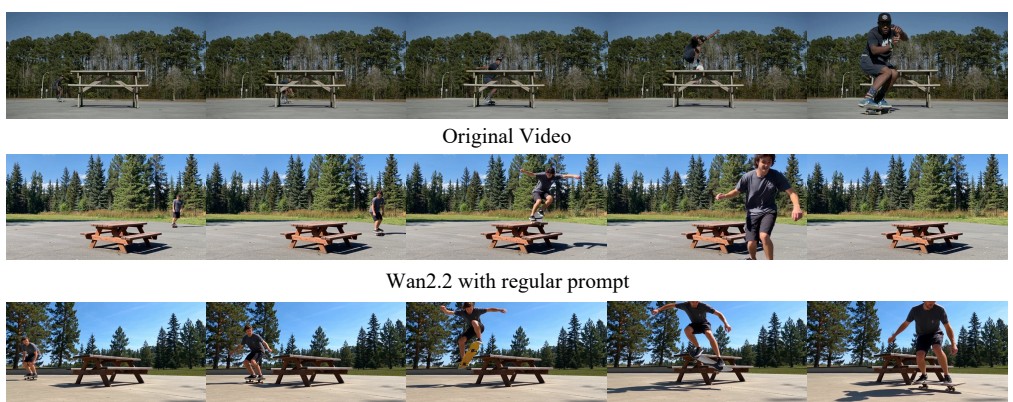

Figure 7: Illustration of finetuning the distilled VLM using the constructed training set. Specifically, we only train the token projector and the Language model decoder during finetuning.

Original Video

Wan2.2 with regular prompt

Wan2.2 with regular prompt + motion extended prompt

Figure 8: Examples of Wan2.2 T2V baseline models tested with regular prompt and with regular prompt plus motion-related extended prompt. The extended version can follow the direction the skateboarder comes better.

## A.2 LABEL TEXT AND MOTION ON TRAINING VIDEO

In order to construct spatiotemporally grounded supervision for video, we extend the static point–text annotations introduced in Appendix A.1 into dynamic trajectories that encode both motion and semantics. The pipeline consists of three major components: (i) representative point selection, (ii) localized text generation, and (iii) temporal propagation via TAP.

**Representative point selection and localized text assignment.** For each video frame, entity masks are first obtained using Grounded SAM (Kirillov et al., 2023; Ravi et al., 2024). To summarize each entity with a small but informative set of anchor points, we apply an adaptive sampling strategy. If the number of foreground pixels in the mask is below a threshold (set to $0.01 \times H \times W$, where $H$ and $W$ are the frame height and width), the entity is represented by a single point: the center of its bounding box. For larger entities, the mask's bounding box is partitioned into a grid of roughly square subregions, such that each subregion covers at most the threshold number of pixels. Within each subregion containing foreground pixels, we compute the bounding box of the local foreground and take its center as the representative point. This ensures that large entities with complex shapes are covered by multiple anchors, while small entities are efficiently represented by a single point. Each representative point is then paired with a coordinate-based query and passed to our point description VLM. This step yields concise, location-specific captions to these sets of representative points.

**Trajectory propagation with TAP.** Once static point to text pairs are obtained on the initial frame, we convert them into full trajectories using Tracking-Any-Point (TAP) (Doersch et al., 2023). TAP is a transformer-based video point tracker capable of following arbitrary query points over long temporal horizons. Given a sampled point $(x_0, y_0)$ on frame 0, TAP predicts its corresponding coordinates $(x_t, y_t)$ across subsequent frames $t$ given a video. Importantly, TAP outputs both tracked positions and visibility flags, where the latter indicate occlusion or out-of-frame states. This allows each trajectory to encode not only motion but also reliability. Every propagated point inherits its associated localized description, producing a trajectory–text pair that maintains semantic grounding over time.

**Final dataset.** The result of this procedure is a collection of temporally consistent trajectories, each annotated with descriptive text. Formally, each unit is represented as a sequence

$$\langle (x_t, y_t, v_t)_{t=0}^T, \ m \rangle, \tag{14}$$

where $(x_t, y_t)$ are pixel coordinates at frame $t$, $v_t \in \{0, 1\}$ denotes visibility, and $m$ is the localized caption. Together, these labeled trajectories provide dense, multimodal supervision for training, capturing both the where and what information of entities throughout the video.

# B  BASELINE IMPLEMENTATION DETAILS

## B.1  WANT2V 2.2

We compare TGT against the Wan2.2 14B text-to-video model Wang et al. (2025a), under two baseline setups:

- Wan2.2 14B text-to-video with **global** prompt only.
- Wan2.2 14B text-to-video with **global + local** prompt.

In both cases, we use the official Wan2.2 T2V 14B model and its released implementation. For the global-only setup, the model is conditioned on the global caption extracted from the original reference video using the Qwen2.5-VL model, which is also the video prompt input to all evaluated methods unless specified otheriwse. For the global + local setup, the model is conditioned on both the global caption and the local prompts, where the latter are obtained using the same distilled VLM employed in our data pipeline. Figure 8 presents an example of generated outputs alongside their corresponding prompts for the Wan2.2 14B T2V baseline. Wan2.2 with extended prompt that describes motion can reconstruct high-level movement in original video better.

## B.2  TORA & MOTIONCTRL

We evaluate TGT against Tora (Zhang et al., 2025) and MotionCtrl (Wang et al., 2024) under a unified protocol. We first use the ground-truth segmentation mask on the initial frame to identify all entities and take the center representative point of each instance. Given these points, we run TAP to extract per-entity 2D trajectories across the sequence. Both Tora and MotionCtrl assume trajectories remain visible throughout; however, TAP marks some timesteps as "invisible" when tracking confidence falls below a threshold. For generation, we ignore TAP's visibility flags and feed the full continuous trajectories to these two methods to satisfy their input assumptions. For evaluation, we report trajectory error metrics only on visible timesteps after temporal alignment, ensuring a fair comparison

## B.3  TRAILBLAZIER

TrailBlazier (Ma et al., 2024b) takes multi-frame bounding boxes with associated local text, builds a representation for each box (subject), and then fuses them into a single video. To form its inputs, we run grounded SAM on the video, guided by the dataset's ground-truth segmentation masks, to obtain per-entity bounding boxes. We derive a short textual description for each box by querying our distilled VLM at the box center on the initial frame to produce a local prompt. Because TrailBlazier requires box locations over multiple frames, we follow its setup and uniformly sample boxes and descriptions at $1/4$ of the sequence length to provide a sparse trajectory of boxes. TrailBlazier then generates a representation for each subject and fuses them to produce the final video.

# C  ADDITIONAL QUALITATIVE RESULTS

All videos are available in the uploaded supplementary materials. Figure 9, Figure 10, and Figure 11 illustrate qualitative results from TGT when conditioned on trajectories with different local prompt inputs. In all three examples, only basic scene-level information is given in the global prompt, without any explicit description of motion or interactions. The additional text-grounded trajectories are therefore responsible for shaping the observed behaviors. In Figure 9, an object is guided to float

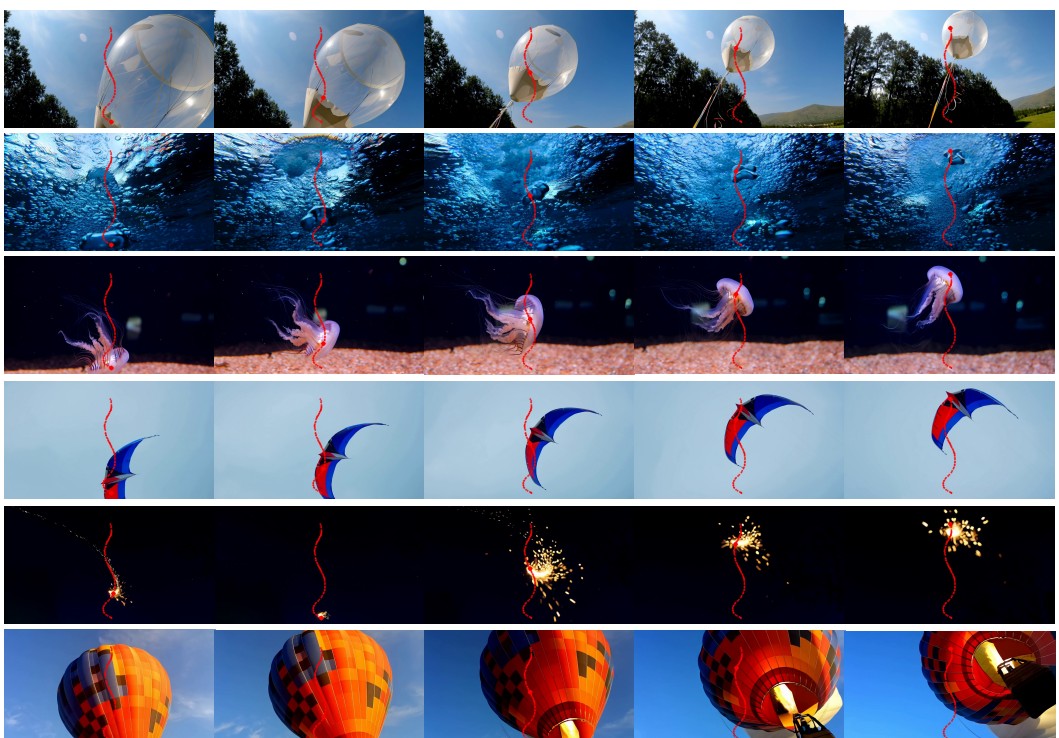

a helium balloon / a stream of bubbles / a jellyfish / a kite / a glowing firefly / a hot-air balloon

Figure 9: Video generation results of TGT with a floating upwards trajectory under different local prompts.

upwards, while Figure 10 shows a subject consistently descending on staircases. Figure 11 demonstrates more complex compositions involving multiple entities, each following its own trajectory. Across these cases, TGT produces correct items and coherent motions that align with the specified texts and trajectories, confirming the effectiveness of our method.

## D LLM USAGE

We used large language models (LLMs) in two limited ways: (i) to help generate and refine example content such as candidate captions/local prompts for qualitative demonstrations, and (ii) to assist with wording, formatting, and editing during manuscript preparation. All model-suggested text and prompts were reviewed, edited, or discarded by the authors; no experimental design, implementation, or quantitative analysis depended on LLM output.

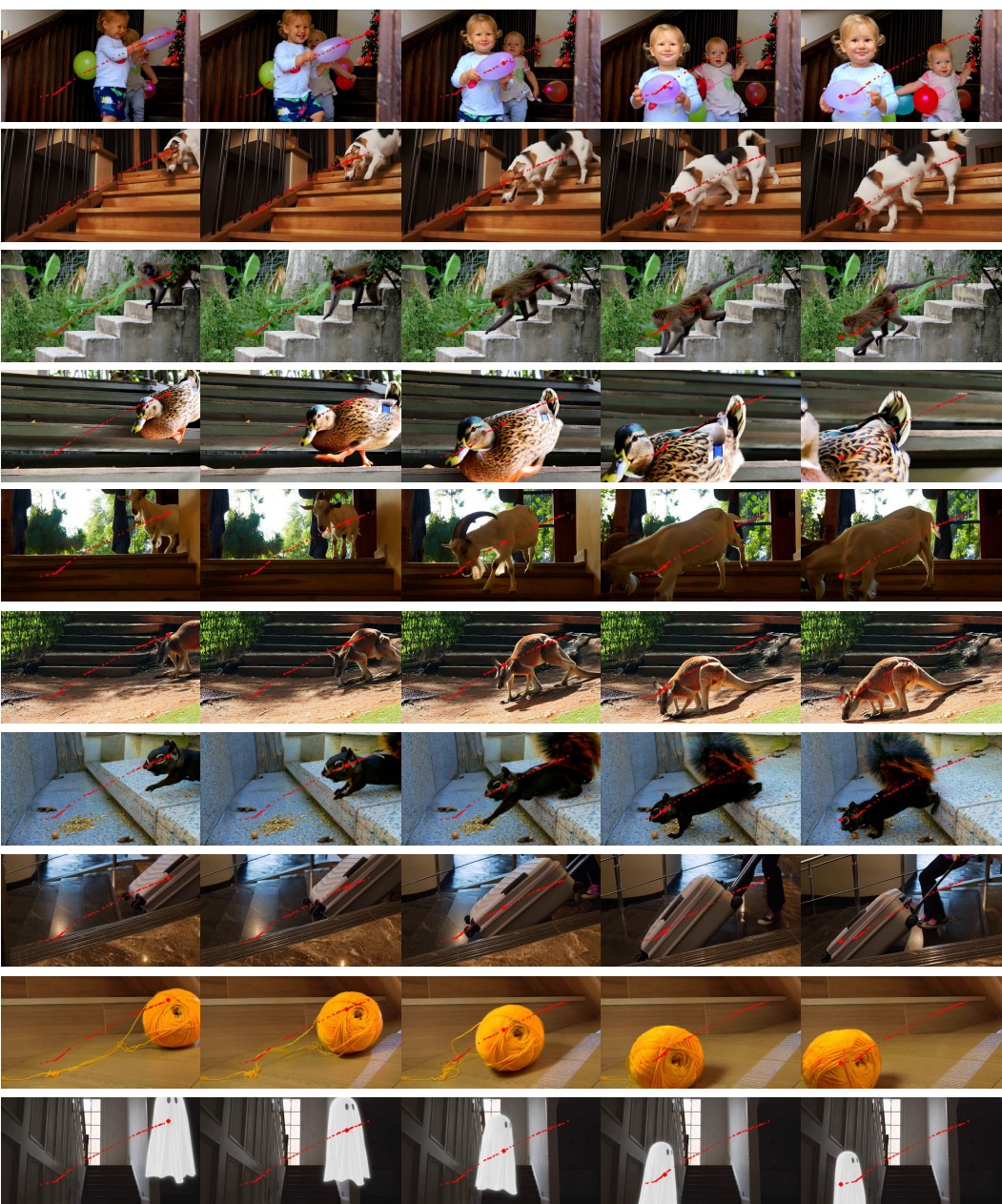

A child with a balloon / A cat / A monkey / A duck / A goat / A kangaroo / A squirrel / A suitcase / A ball of yarn / A ghost

Figure 10: Video generation results of TGT with a walking-down-stairs trajectory under different local prompts.

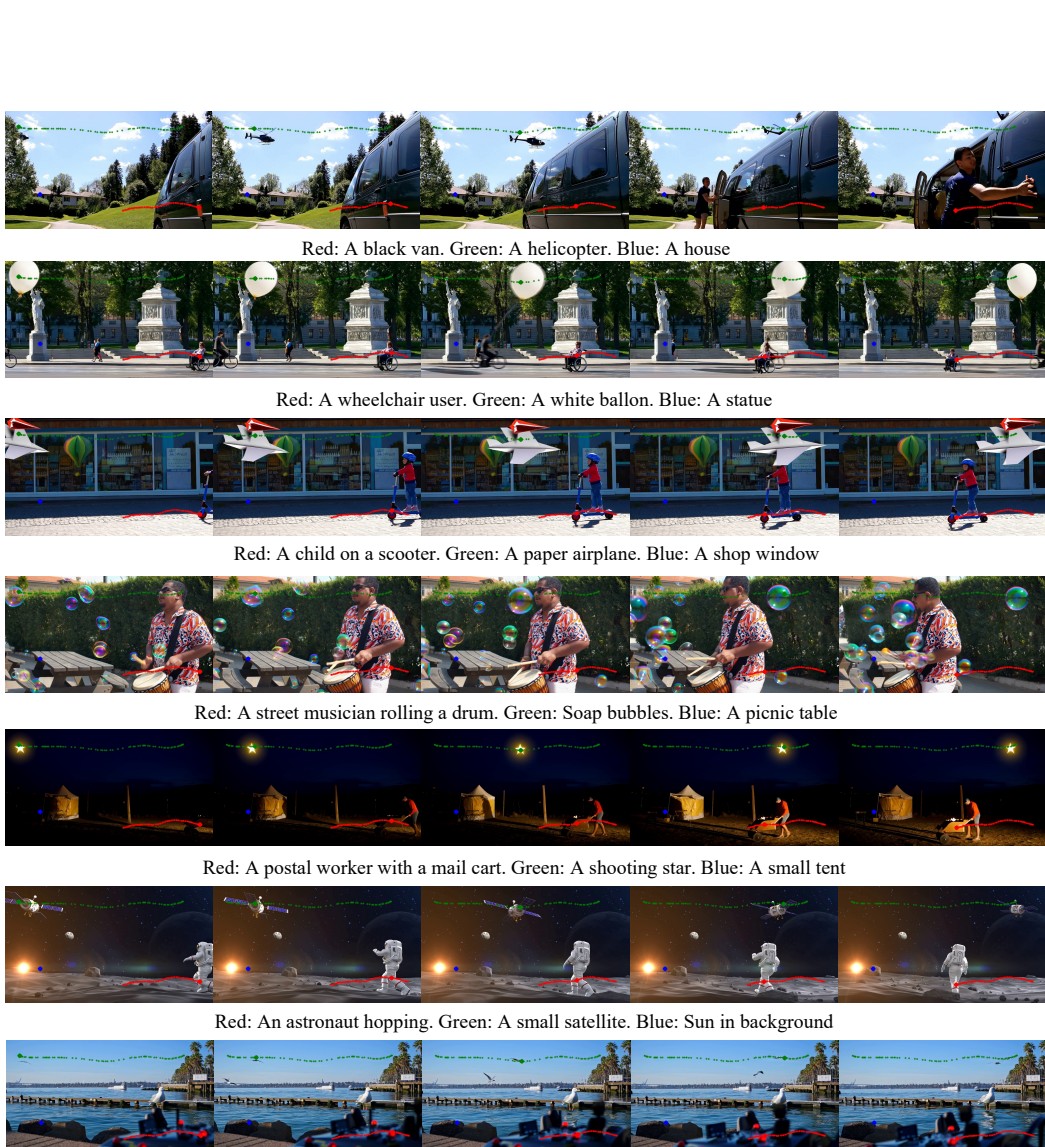

Red: A black van. Green: A helicopter. Blue: A house

Red: A wheelchair user. Green: A white ballon. Blue: A statue

Red: A child on a scooter. Green: A paper airplane. Blue: A shop window

Red: A street musician rolling a drum. Green: Soap bubbles. Blue: A picnic table

Red: A postal worker with a mail cart. Green: A shooting star. Blue: A small tent

Red: An astronaut hopping. Green: A small satellite. Blue: Sun in background

Red: A remote-control car. Green: A seagull. Blue: A rock in sea

Figure 11: Video generation results of TGT with multiple trajectories under different local prompts.

