# OpenReview forum: "TGT: Text-Grounded Trajectories for Locally Controlled Video Generation"
_ICLR.cc/2026/Conference — ICLR 2026 Conference Withdrawn Submission_

### Official Review · Reviewer_bcpG · 2025-10-30

**Soundness:** 3
**Presentation:** 3
**Contribution:** 2
**Rating:** 6
**Confidence:** 4

**Summary:**

This paper presents Text-Grounded Trajectories (TGT), a framework for controllable text-to-video generation that links localized text descriptions with motion trajectories. The key innovation lies in the Location-Aware Cross-Attention (LACA) module, which effectively fuses spatial trajectory cues with textual semantics, and a dual-CFG strategy that decouples local (object-level) and global (scene-level) guidance.

**Strengths:**

- This paper has strong motivation, proposing the first paradigm that guide point-based trajectory controllable video generation with per-trajectory text prompts.

**Weaknesses:**

- Abuse of notation: In Section 3.1, $L$ is used to denote the length of the encoded text prompt and $N$ is used to denote the token number of the video in latent space. However, in Section 3.2, $L$ is used to denote the token number of the pachified video in latent space and the length of the encoded local text prompt. This is confusing and it needs explanation that why the pachified video and local text prompt have the same token number.
- Uninformative and inconsistent figures: Figure 2 is uninformative as it does not provide a clear overview of the proposed pipeline. What does the right half of Figure 2 mean? Does it only means that the video latent is flattened in spatial and temporal dimensions? And it is inconsistent with the proposed method since the LACA module incoporates both the local text prompt and global text prompt, but Figure 2 only shows that the LACA module takes the local text prompt as input.
- Unknown metrics: In the experiments, the authors use EPE while there is no explanation nor reference for it. The authors need to clarify what EPE is and how it is calculated.
- Missing experiments result: The table 2 only shows the results of baselines and the results of the proposed method are missing.

**Questions:**

See Weaknesses

---

### Official Review · Reviewer_Agjd · 2025-10-30

**Soundness:** 3
**Presentation:** 3
**Contribution:** 2
**Rating:** 6
**Confidence:** 4

**Summary:**

This paper introduces Text-Grounded Trajectories (TGT) — a novel framework for locally controllable text-to-video generation that pairs motion trajectories with localized textual descriptions. The goal is to achieve precise spatial and temporal control in generated videos while maintaining high visual quality and semantic consistency.

**Strengths:**

1. The paper addresses a clear and important limitation in current text-to-video (T2V) generation — the lack of fine-grained spatial and motion control. By introducing text-grounded trajectories as a means to link motion trajectories with localized textual descriptions, the work targets a gap that is both technically relevant and of high practical importance for controllable video generation.
  2. The proposed LACA module represents a conceptually elegant and technically lightweight addition to the DiT-based backbone. Its ability to inject local semantic information into the diffusion process using Gaussian-weighted attention around trajectory points is both intuitive and effective. The design maintains modularity and compatibility with pretrained large-scale video generation models, which enhances the method’s practicality.
  3. The dual-CFG strategy is an insightful design choice that decouples global and local guidance strengths. It allows for a flexible trade-off between overall prompt fidelity and motion precision — a feature that has often been lacking in prior controllable video generation frameworks.

**Weaknesses:**

1. While the method demonstrates improved spatial controllability, the evaluation largely focuses on short clips (≈80 frames). The paper would benefit from a systematic analysis of temporal consistency, particularly under longer sequences or occlusions. It remains unclear whether TGT maintains stable motion trajectories over extended time horizons.
  2. The qualitative results (Figures 4–11) are visually impressive but cover a relatively narrow range of scenarios — mainly single or few-object cases with simple backgrounds. Complex real-world scenes (e.g., crowd motion, camera movement, or physical interactions) are not thoroughly tested, limiting claims about scalability and robustness.
  3. While TGT presents a well-integrated framework, much of its novelty lies in combining existing components — such as trajectory-based control, localized text conditioning, and DiT backbones — rather than introducing a fundamentally new modeling paradigm. The LACA module, though elegant, can be seen as an incremental extension of existing cross-attention mechanisms (e.g., spatially masked or region-conditioned attention).

**Questions:**

1. Generalization and Scalability. How well does TGT generalize to unseen object categories or scene types that were not included in the localized-text training dataset? Could the authors comment on how the method scales with the number of trajectories (e.g., more than 10 objects simultaneously controlled)?
  2. Ablation on Guidance Scales and Hyperparameters. The dual-CFG setup introduces two independent scales, sglobs_{glob}sglob and slocs_{loc}sloc. How sensitive is model performance to their specific choice (5 and 4)? Would adaptive or learned scaling mechanisms perform better than fixed values?
  3. Temporal Robustness. Can TGT maintain coherent trajectories over long videos (>200 frames) or under partial occlusions? How does the model handle situations where tracked points temporarily disappear or reappear?

---

### Official Review · Reviewer_7VsE · 2025-10-31

**Soundness:** 2
**Presentation:** 2
**Contribution:** 2
**Rating:** 4
**Confidence:** 3

**Summary:**

The paper introduces Text‑Grounded Trajectories (TGT), a controllable text‑to‑video (T2V) framework that ties sparse 2D point trajectories to localized text prompts, enabling joint control of what moves (appearance/identity) and how/where it moves (trajectory). Architecturally, the method adds a lightweight Location‑Aware Cross‑Attention (LACA) branch to each DiT block and uses a dual classifier‑free guidance (dual‑CFG) scheme with separate scales for the global caption and the local text. Because no dataset exists with point‑trajectory ↔ local‑text pairs, the authors build a two‑stage data pipeline: (i) use Grounded‑SAM to segment a key frame, choose representative points, and distill a coordinate‑aware local captioner by prompting a teacher VLM (GPT‑4o) and fine‑tuning Qwen2.5‑VL‑3B; (ii) propagate points with TAP to get long trajectories and visibility flags. Empirically, TGT achieves the lowest EPE and the highest local CLIP‑T, outperforming WanT2V, MotionCtrl, TrailBlazer, and Tora.

**Strengths:**

1. Strong empirical results compared with previous SoTA T2V generators across multiple evaluation metrics (e.g., EPE, CLIP-T).

2. Detailed ablations demonstrate the effectiveness of each component in the model (e.g., CFG design, LACA component).

3. Potential applications to other tasks like video-to-video generation / video editing tasks.

**Weaknesses:**

1. Utilizing attention for better grounding with the prompt has been explored in many previous approaches (e.g., Video-Editing-Attention[1], DreamRunner[2], Presto[3]). Though these approaches are not directly for trajectory control, but they can be easily extended for trajectory control.

2. Evaluation is only on the curated dataset. It's worth checking the performance on how this approach can be extended to general T2V generation (e.g., PhyGenBench[4], T2VCompBench[5]), where these benchmarks also contain some examples that actually can be done with trajectory control (e.g., motion class in T2VCompBench).

[1] Investigating the Effectiveness of Cross-Attention to Unlock Zero-Shot Editing of Text-to-Video Diffusion Models

[2] DreamRunner: Fine-Grained Compositional Story-to-Video Generation with Retrieval-Augmented Motion Adaptation

[3] Long Video Diffusion Generation with Segmented Cross-Attention and Content-Rich Video Data Curation

[4] Towards World Simulator: Crafting Physical Commonsense-Based Benchmark for Video Generation

[5] T2V-CompBench: A Comprehensive Benchmark for Compositional Text-to-video Generation

**Questions:**

N/A

---

### Official Review · Reviewer_TNvT · 2025-11-01

**Soundness:** 3
**Presentation:** 3
**Contribution:** 2
**Rating:** 4
**Confidence:** 4

**Summary:**

This paper proposes TGT (Text-Grounded Trajectories), a framework for controllable text-to-video generation that combines point-based trajectories with localized text descriptions. It introduces a lightweight Location-Aware Cross-Attention (LACA) module and a dual classifier-free guidance (dual-CFG) scheme to separately control motion and appearance via global and local prompts. To support training, the authors also construct a large-scale dataset by automatically annotating trajectories with localized captions. Experimental results on standard benchmarks demonstrate that TGT achieves stronger motion controllability and text alignment than existing baselines, with improved visual quality and flexibility in applications such as video editing.

**Strengths:**

- The paper is clearly written and well-structured, with a good motivation and comprehensive methodology.

- LACA and Dual CFG designs are intuitive and proven to be effective.

- The experimental section is thorough.

- The proposed approach demonstrates good results, outperforming prior methods in both motion control and alignment with local text.

- The framework is practically useful, enabling applications such as video-to-video mirroring and localized text-driven editing.

**Weaknesses:**

**Overstatement of the Problem Framing and Contribution Gap**

The paper frames its main contribution as introducing explicit text-grounded trajectories in text-to-video generation, suggesting this has not been addressed in prior literature. However, this characterization overlooks a range of recent works that already leverage trajectory-like spatial control in close association with local text descriptions. For example, PEEKABOO [1], BlobGEN-Vid [2], CineMaster [3], and 3DTrajMaster [4] all incorporate entity-specific prompts that are tied to either 2D or 3D trajectory representations such as blobs, layout paths, or pose streams. Therefore, the novelty of the “text-to-trajectory grounding” problem, as presented, is a bit overstated.

**Limited Technical Novelty in Model Design**

The core technical component, Location-Aware Cross-Attention, amounts to spatially masking attention based on trajectory proximity and injecting local text features into those neighborhoods. While effective, this is a relatively standard design pattern in compositional generation models. Similar masked or region-focused local cross attention designs have already been used in Videotetris [5], DreamRunner [6], PEEKABOO [1], BlobGEN-Vid [2], etc, for grounding local texts to specific spatial-temporal regions.

The dual CFG strategy also closely mirrors prior work such as InstructPix2Pix [7], where independent CFG weights are applied to different input modalities (e.g., source image vs. edit prompt). TGT adapts this paradigm to global and local text, but the structural idea and formulation remain unchanged. As such, both the attention design and the guidance scheme may represent incremental improvements rather than conceptual innovations.

**Missing Baseline for Data Pipeline**

The paper presents the data pipeline as a contribution. While interesting,  it is not compared with a straightforward baseline that uses existing trajectory annotation methods (that have already been introduced in many video trajectory control papers) with an additional captioning step. With modern tools like SAM2 and image captioning models, assigning a localized caption to each trajectory is relatively easy. A simple pipeline using “SAM2 + caption model” should be included as a baseline. Without such a comparison, it remains unclear whether the proposed pipeline offers any meaningful advantage.


---
[1] PEEKABOO: Interactive Video Generation via Masked-Diffusion, CVPR 2024

[2] BlobGEN-Vid: Compositional Text-to-Video Generation with Blob Video Representations, CVPR 2025

[3] CineMaster: A 3D-Aware and Controllable Framework for Cinematic Text-to-Video Generation, SIGGRAPH 2025

[4] 3DTrajMaster: Mastering 3D Trajectory for Multi-Entity Motion in Video Generation, ICLR 2025

[5] Videotetris: Towards Compositional Text-to-Video Generation, NeurIPS 2024

[6] DreamRunner: Fine-Grained Compositional Story-to-Video Generation with Retrieval-Augmented Motion Adaptation, 2024

[7] InstructPix2Pix: Learning to Follow Image Editing Instructions, CVPR 2023

**Questions:**

See weakness

---

### Note · Authors · 2025-11-14

I have read and agree with the venue's withdrawal policy on behalf of myself and my co-authors.